# CD46 and Oncologic Interactions: Friendly Fire against Cancer

**DOI:** 10.3390/antib9040059

**Published:** 2020-11-02

**Authors:** Michelle Elvington, M. Kathryn Liszewski, John P. Atkinson

**Affiliations:** 1Kypha, Inc., Saint Louis, MO 63110, USA; melvington@kypha.net; 2Division of Rheumatology, Department of Medicine, Washington University School of Medicine, Saint Louis, MO 63110, USA; kliszews@wustl.edu

**Keywords:** CD46, membrane cofactor protein (MCP), complement, cancer therapeutics, measles virus, adenovirus, antibody-drug conjugates

## Abstract

One of the most challenging aspects of cancer therapeutics is target selection. Recently, CD46 (membrane cofactor protein; MCP) has emerged as a key player in both malignant transformation as well as in cancer treatments. Normally a regulator of complement activation, CD46 is co-expressed as four predominant isoforms on almost all cell types. CD46 is highly overexpressed on a variety of human tumor cells. Clinical and experimental data support an association between increased CD46 expression and malignant transformation and metastasizing potential. Further, CD46 is a newly discovered driver of metabolic processes and plays a role in the intracellular complement system (complosome). CD46 is also known as a pathogen magnet due to its role as a receptor for numerous microbes, including several species of measles virus and adenoviruses. Strains of these two viruses have been exploited as vectors for the therapeutic development of oncolytic agents targeting CD46. In addition, monoclonal antibody-drug conjugates against CD46 also are being clinically evaluated. As a result, there are multiple early-phase clinical trials targeting CD46 to treat a variety of cancers. Here, we review CD46 relative to these oncologic connections.

## 1. Introduction to CD46

CD46 is a ubiquitously expressed membrane protein in humans that regulates C3b/C4b that deposits on host cells and serves as a receptor for multiple pathogens. CD46 was originally discovered as a C3b-interacting protein based on its ability in surface-labeled, solubilized cell preparations to bind to C3b and C3(H_2_O) affinity columns [1]. Subsequently, it was demonstrated to be a cofactor protein for the plasma serine protease factor I (FI) to mediate cleavage of C3b to iC3b and C4b to C4c and C4d [2,3]. Further, CD46 performs its function intrinsically; that is, this inhibitor binds its two ligands, C3b and C4b, only if they are deposited on the same cell on which CD46 is expressed. As a membrane protein, CD46 does not bind to C3b or C4b free in blood, attached to immune complexes or bound to other cells. CD46 has a wide tissue distribution, being expressed by almost all human cell populations except erythrocytes [3,4]. Thus, CD46 protects healthy human cells from complement attack.

CD46 is rather unique among complement proteins, receptors and regulators in that most cell types co-express four primary isoforms (Figure 1). These arise from alternative splicing of a single gene (Figure 2) located within the regulators of complement activation (RCA) cluster on the long arm of chromosome 1 at position q3.2 [5]. The N-terminus of CD46 consists of four contiguous complement control protein (CCP) repeats. Following this is an alternatively spliced segment enriched in serines, threonines and prolines (i.e., the STP domain) that is variably *O*-glycosylated. The STP domain of the regularly expressed proteins consists of B+C or C alone, although rarer isoforms exist. Following the STP domain and common to all isoforms is a juxtamembraneous segment of 12 amino acids of undefined function and a hydrophobic transmembrane domain. CD46 also contains one of two cytoplasmic tails (termed CYT-1 or CYT-2), each of which has distinct signaling motifs (reviewed in [3]). Much remains to be learned about the biological reasons for CD46 isoform variation among cell types.

Since its original discovery in 1987, CD46 is now known to play a broader role in human biology (reviewed in [6,7]). This includes modulation of immune cell function through its intracellular signaling capabilities; namely, CD46 is an important modulator of adaptive immunity and a central participant in human infectious and Th1 biology [8,9]. Individuals with a homozygous CD46 deficiency are hampered in mounting Th1 responses and suffer from recurrent severe infections [10] and, more commonly, atypical hemolytic uremic syndrome at a young age [11]. CD46 engagement on CD4^+^ T cells results in a Th1 response; however, as IL-2 accumulates, a switch occurs to a T regulatory phenotype characterized by production of IL-10 that is mediated by the CYT-1 tail of CD46 [12,13]. In this manner, CD46 also contributes to the contraction phase of a Th1 response. CD46 cytoplasmic tail switching links Th1 cell activation (CYT-1) and then contraction (CYT-2) to a pathway for metabolic reprogramming [14]. The bulk of the work on this subject has been performed by the Kemper laboratory employing human CD4^+^ T cells and is reviewed extensively elsewhere [6,8,15].

Four additional key observations relevant to this review are CD46′s: (1) enhanced expression by many types of malignant cell populations; (2) “abuse” by ten human-specific pathogens that use CD46 as a receptor, including species of measles virus and adenovirus; (3) modulation of immune cell function via its intracellular signaling capabilities; (4) role as a driver of cellular metabolism [16]. Point #3 is briefly noted in the preceding paragraph while the other three points will be further highlighted in this review because of their oncologic implications.

## 2. CD46 as Tumor Target

CD46 is emerging as an important player in both malignant transformation as well as in cancer immunotherapy. Complement expression is often dysregulated in tumors [17] (reviewed in [18]). Although heterogeneous relative to tumor type, complement proteins expressed by malignant cells are variable; in general, tumor cells express low C8 and C9 but highly express C3 and the regulators factor H, factor I and especially the membrane regulators CD46, CD55 and CD59 [19]. Notably, CD46 expression is increased up to 14-fold in relapsed multiple myeloma (MM) patients who have the region on chromosome 1q carrying CD46 genomically amplified (mean antigen density 313,190 for MM vs. 22,475 for healthy donor plasma cells) [20]. The overexpression of CD46 occurs in many, although not all, other common tumor types but usually to a lesser magnitude. For example, in hepatocellular carcinoma cells, the relative density of CD46 is increased approximately 6-fold [21]. Complement deposition also is frequently noted in tumor tissue and soluble activation fragments are identified in patients’ sera [22,23], including increased levels of soluble CD46 [24]. Further, clinical and experimental data in ovarian and breast cancer as well as multiple myeloma support an association between increased CD46 expression and malignant transformation and metastasizing potential [25,26,27]. Indeed, increased CD46 expression is a prognostic indicator in multiple common malignancies, including ovarian cancer, breast cancer and hepatocellular carcinoma [25,26,28].

In the context of cancer, the view of the complement system has traditionally been that of an anti-tumor effector, particularly to enhance the efficacy of anti-cancer monoclonal antibodies (mAb) [29,30]. Complement engagement following Ab binding to a tumor Ag has several potential mechanisms to enhance cell killing [31]. These include direct complement-mediated lysis or complement-dependent cytotoxicity (CDC) and enhancing antibody-dependent cellular cytotoxicity (ADCC). Additionally, C3a and C5a can recruit immune cells that then may modulate the adaptive immune response. The upregulation of complement inhibitors, including CD46, has historically been considered an evasion mechanism against Ab therapy that may also prevent the generation of an acquired immune response (reviewed in [17]). The interference of membrane complement regulators with the efficacy of anti-cancer mAb therapy has been demonstrated in animal models ([32] and reviewed in [33,34]). Several strategies have been employed to target the overexpression of complement regulators and thereby improve the therapeutic outcome [34]. These include small interfering RNAs [35,36,37], neutralizing (blocking) mAbs to complement inhibitors [38,39], bispecific antibodies [40,41,42,43] and a targeted complement activator (CR2-Fc) [30,44].

Targeted downregulation of the in vivo expression of a complement inhibitor, such as CD46, is technically challenging due to its ubiquitous presence on normal cells. However, the feasibility of this approach has been demonstrated both in vitro and in animal models where tumor cells can be manipulated ex vivo. For example, siRNA-mediated downregulation of CD46, CD55 and/or CD59 on several primary tumors and tumor cell lines sensitizes them to CDC [35,36]. Furthermore, the downregulation of the complement inhibitor Crry (the murine counterpart of CD46) on tumor cells in a mouse model of metastatic bladder cancer induced a protective anti-tumor CD8^+^ T cell response [37]. In another model, complement inhibitor neutralization on tumor cells prior to vaccination elicited protective immunity [38]. Proof of principle also has been demonstrated in murine models for the efficacy of neutralizing mAbs [39] and bispecific antibodies [40,41,42,43]. Another study determined that in primary myeloma cells derived from bone marrow aspirates, a macropinocytosing CD46-antibody drug conjugate induced apoptosis and cell death but did not affect the viability of nontumor mononuclear cells [20]. In a slightly different approach, a strategy to utilize the targeting specificity of a mAb to amplify complement activation on tumor cells (CR2-Fc) demonstrated efficacy in murine models [30,44].

Unfortunately, complement activation does not always act to the host’s benefit [45,46]. The protumor roles of complement were discussed recently in an excellent review [19]. C5a, in particular, has been shown to be protumor by recruitment of myeloid-derived suppressor cells and T regulatory cells that suppress anti-tumor T cell responses [47,48]. Complement may also promote tumorigenesis by triggering tumor cell proliferation, invasiveness and metastasis, as well as by enhancing angiogenesis [49]. Therefore, although increased CD46 on tumor cells is generally considered an immune evasion mechanism, in certain situations, preventing complement activation may also benefit the host. For example, suppressing complement activation at the level of C3 will also inhibit downstream C5 cleavage and generation of C5a, which has been shown to have protumor effects [45,47]. Another possibility is that the role of complement in promoting the clearance of apoptotic cells, which is a generally tolerogenic process, supports tumor growth. Indeed, apoptotic cells have been shown to directly activate complement [50,51,52]. This possibility was demonstrated in a murine model of lymphoma in which delivering a targeted complement inhibitor enhanced the outcome of radiation therapy by inhibiting apoptotic cell clearance, thereby promoting inflammation within the tumor and driving a systemic antitumor immune response [53].

Notably, a body of literature, as well as online resources such as The Cancer Genome Atlas (TCGA) Program reveal that the involvement of CD46 in cancer varies. CD46 expression in malignancies may: (1) correlate with a poorer prognosis (e.g., breast, colorectal, prostate and cervical cancers); (2) correlate with a positive outcome (e.g., renal or stomach cancers); (3) have no apparent impact. Further, there can be variability in expression even among specific tumor cell types. For example, one study reported that 35% of patients with colon or prostate carcinoma overexpressed CD46, while in other cancers (such as brain, lymphoma and lung), ~11% of tissue samples overexpressed CD46 [54]. Such variability suggests differences in the potential mode of action of CD46 in different contexts. Dissecting the role CD46 plays in multiple settings will support which patient cohorts would benefit from specific CD46-targeted therapeutics.

## 3. CD46 as a Receptor for Anti-Tumor Therapeutic Vectors

Interestingly, CD46 is known as a “pathogen magnet” because it is a target for ten (and counting) human pathogens [55,56,57,58], including the Edmonston vaccine strain of the measles virus [59] and some species of adenovirus (see below). Moreover, bovine and swine CD46 are receptors for the bovine viral diarrhea virus and the classical swine fever virus, respectively [60,61]. CD46 is utilized as a receptor for pathogens to gain entry into the cell. The exact reason why so many microbes target CD46 is unclear. However, several issues likely drive this phenomenon, including its high level of and ubiquitous expression across human cell types. In addition, because pathogens bind to different sites on the protein (see Figure 3) and utilize distinct methods for cellular entry, CD46 is likely targeted not simply due to its wide distribution and high level of expression but also because of its immune-modulatory signaling functions including immune suppression [56]. For example, *Neisseria gonorrhoeae* dysregulates the CD46-CYT-1 autophagic pathway to disturb lysosome homeostasis and promote its survival [62]. Furthermore, measles virus downregulates IL-12 production by monocytes through its binding to CD46 [63,64].

One of the most challenging aspects of cancer therapy development is target selection. In this regard, two features make CD46 an attractive candidate. First, as noted in the preceding paragraph, it is a receptor for some strains of measles- and adenoviruses. Second, as outlined above, CD46 is commonly overexpressed on a variety of tumor cells [27,33]. Thus, a number of therapeutic agents, including more than 20 now in clinical trials (Table 1), employ CD46-targeted oncolytic adenoviral- or measles virus-based vectors [20,54,65,66,67,68]. More than 500 gene therapy trials have been conducted, most with human adenoviral vectors. Many have been conducted using Adenovirus type C that binds to the coxsackie-adenovirus receptor (CAR) (reviewed by [69]). However, the low expression level of CAR by cancer cells has limited the efficiency of such targeted oncolytic therapies. Alternatively, some species B adenoviruses target CD46, thus presenting an improved approach to increase the effectiveness of cancer targeting. As an example, comparisons between CD46 and CAR-targeted adenoviral therapy have demonstrated superior outcomes with CD46 targeting in bladder cancer [69] and ovarian cancer [70], despite comparable receptor expression by the tumor cells.

Other oncolytic strategies utilize measles virus that also targets CD46. For example, the Edmonston lineage (vaccine) strain of measles virus has been modified to express carcinoembryonic antigen or the human sodium iodide symporter (NIS) for noninvasive monitoring. These are being tested in a variety of tumor types (reviewed in [71] and see Table 1). Notably, treatment of an MM patient with an engineered oncolytic measles virus containing the NIS (termed MV-NIS) resulted in durable remission in a case report [72] and demonstrated activity in phase 1 trials for patients with ovarian cancer [73,74] and MM [75].

Another CD46-targeted therapy employs an antibody-drug conjugate (ADC) [20]. In this case, the CD46-targeting specificity is mediated by an Ab instead of a virus. As noted above, the region on chromosome 1q carrying CD46 commonly undergoes genomic amplification in relapsed myeloma patients. This amplification correlates with markedly increased expression of CD46 by the tumor [20]. The potential of the CD46-ADC approach was demonstrated in MM in which potent inhibition of myeloma cell proliferation was achieved in an orthostatic xenograft (mouse) model [20]. The CD46-ADC approach has also shown efficacy in metastatic castration-resistant prostate cancer, utilizing a tubulin inhibitor conjugated as the ADC added to a macropinocytosing anti-CD46 antibody (Ab) [65]. These studies support CD46 as a promising target for antibody-based therapeutics for certain tumor types. Two phase 1 trials using an anti-CD46 ADC are being conducted against MM and metastatic prostate cancer (Table 1).

Factors affecting the expression of CD46 and loss from the cell surface also need to be taken into consideration relative to targeting of CD46 as a therapeutic strategy. For example, a recent report demonstrated that p53, which is a frequently deleted or modified gene in human cancers, impacts CD46 expression and, thus, MM susceptibility to oncolytic measles virus [76]. Further, CD46 is known to be internalized by two mechanisms. CD46 can be constitutively internalized via clathrin-dependent endocytosis and recycled to the surface. In addition, crosslinking of CD46 on the cell surface by multivalent anti-CD46 Ab or measles virus induces ligand-engulfing pseudopodia (a process similar to macropinocytosis) and subsequently leads to CD46 degradation and downregulation on the cell surface [77]. This was demonstrated in vitro, as well as in in vivo murine and non-human primate (NHP) models. In NHPs, CD46 depletion enhanced the depletion of CD20^+^ B cells by a low dose of rituximab [78].

## 4. CD46 as a Metabolic Driver

New understandings of the complement system as well as CD46 are expanding their roles beyond that traditionally appreciated. It has become increasingly recognized that most cells contain an intracellular complement system, ICS or complosome, that not only provides immune defense but also assists in key interactions for host cell functions [79]. For example, autocrine activation of CD46 via C3b plays a critical role in nutrient uptake and enhances cellular metabolism [14]. While the arsenal of complement components and specific cell types constituting the complosome continues to be elucidated, the finding of CD46 as a metabolic driver suggests a possible role in malignant transformation and/or cell proliferation.

Cancer cells are highly glycolytic and preferentially metabolize glucose anaerobically ([80] reviewed in [81]). Interestingly, new connections have been discovered between glucose metabolism and the ICS that indicate that this metabolic pathway may be driven by intracellular C3 signaling [81]. In the ICS of human CD4^+^ T cells, C3a generated intracellularly interacts with its receptor, C3aR, expressed on lysosomes and thus provides tonic low-level activation of the mammalian target of rapamycin (mTOR) signaling pathway [16] (Figure 4). This interaction is an essential survival signal for CD4^+^ T cells. Intracellular complement activity also drives anti-apoptotic activity through intracellularly generated C3b engagement of CD46 on the surface of T cells that subsequently enhances glycolysis [14] (Figure 4). Analogously, CD8^+^ T cells rely on CD46 stimulation of fatty acid metabolism for optimal cytotoxic function [82]. King et al. demonstrated that intracellular C3 in pancreatic islet β cells regulates autophagy to protect β cells from death [83]. Further, CD46 has been shown to interact with Jagged1, a Notch family member, to regulate Th1 cell activation [10]. Notch is known to regulate oxidative phosphorylation and glycolysis in cancer cells [14]. Thus, a new view is emerging that intracellular complement activation regulates cell survival pathways (reviewed in [81]). Future studies addressing CD46-driven metabolic reprogramming are warranted to address its oncologic implications in cancer.

Given the importance of the CD46-C3 axis for cell survival, the source of intracellular C3 has been examined. We have shown that intracellular C3 arises via two distinct routes: either being synthesized in situ [16] or taken up from the extracellular milieu [84]. Specifically, the form taken up from the extracellular fluid is C3(H_2_O). Thus, C3(H_2_O) has two main functions—in the extracellular space, it is a trigger for the AP, and in the intracellular space, it provides a source of C3 for the complosome. In the context of cancer, C3(H_2_O) has not been evaluated, although, notably, dysregulation of the alternative pathway of complement activation, which can be initiated by C3(H_2_O), has been demonstrated in several hematological cancers [85].

Oncogenesis also may be driven by aberrant signaling processes. Buettner et al. correlated overexpression of CD46 mRNA and protein with the binding of activated signal transducers and activators of transcription 3 (STAT3) to two sites in the CD46 promoter [86]. Since STAT3 is persistently activated in a wide variety of tumors and IL-6 is increased with co-stimulation of CD46 in the presence of C3(H_2_O) [84], the CD46-STAT3-C3 axis may play a key role in regulating metabolic and regenerative processes. We hypothesize that CD46 isoform cytoplasmic tail switching (as occurs in activated versus quiescent T cells [87]) may be altered in cancers, such as MM. The potential for CD46-targeting to modulate this axis deserves further investigation.

## 5. Conclusions

Since the discovery of CD46 more than 30 years ago, much knowledge has been gleaned about this multifaceted protein [7]. The surprising finding of its alternative splicing to produce four common isoforms that co-exist on most cells has proven significant in both its function and importance to biology. Much remains to be leveraged about these roles. Because of its widespread expression, it is not surprising that at least ten human-specific pathogens have targeted CD46. Beyond its role in complement regulation, we now also appreciate CD46 as a driver of cellular metabolism and an important player in the newly described intracellular complement system. Collectively, these studies provide a strong rationale for further characterization of these interconnected players in oncology. The success of pre-clinical studies utilizing CD46 as a therapeutic target have now led more than 20 clinical trials being conducted for cancer treatments. Thus, ‘friendly fire’ aimed at CD46 utilizing approaches such as oncolytic viruses or antibody-drug conjugates (alone or in combination with other therapeutics) offers promising new modalities for the treatment of malignancies.

## Figures and Tables

**Figure 1 antibodies-09-00059-f001:**
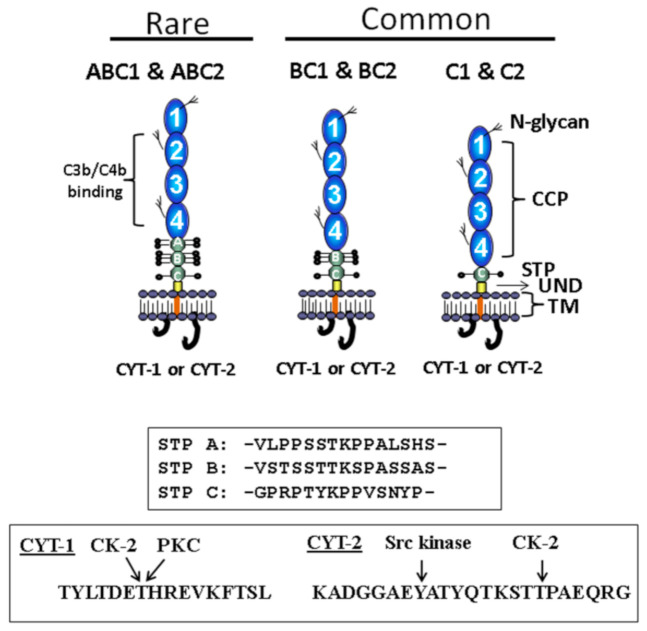
CD46 (membrane cofactor protein; MCP) is a widely expressed, alternatively spliced complement regulatory protein whose structure is dominated by four complement control protein modules (CCPs). CCPs 2–4 house the main sites for C3b and C4b interactions. CCPs 1, 2 and 4 feature *N*-glycans. Alternative splicing of the STP region (enriched in serines, threonines and prolines, and a site of variable levels for *O*-glycosylation) and of the cytoplasmic tails (CYT-1 or CYT-2, which have distinct signaling motifs) generates four common isoforms co-expressed to variable levels on most cells. These are called BC1, BC2, C1 and C2. Rarer isoforms also exist. UND, undefined juxtamembraneous segment; TM, transmembrane domain; CK-2, casein kinase 2; PKC, protein kinase C.

**Figure 2 antibodies-09-00059-f002:**
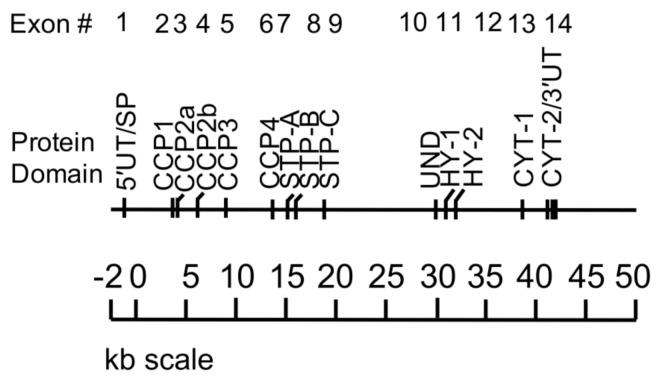
The gene for CD46. The alternatively spliced *MCP* gene lies at 1q3.2 and consists of 14 exons and 13 introns for a minimum length of 43 Kb. The exon number and corresponding domain are indicated. Note that CCP2 and the hydrophobic domain (HY) are encoded by two exons. Abbreviations per Figure 1: 5′UT/SP (5′ untranslated area and signal peptide); CYT-2/3′UT (cytoplasmic tail 2 and 3′ untranslated region).

**Figure 3 antibodies-09-00059-f003:**
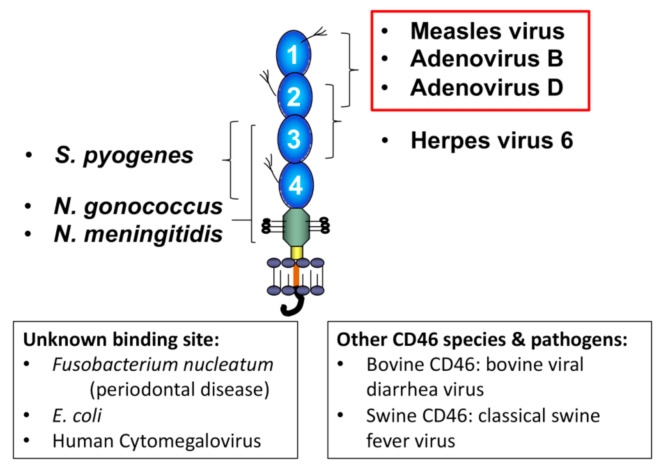
CD46 is a receptor for 10 human-specific pathogens; bovine and swine CD46 are also pathogen receptors. For seven of the human pathogens, the binding site has been identified (indicated above), including strains of measles virus and adenovirus. Two features make CD46 an attractive target for oncology applications: (1) it is overexpressed on many cancer cell types and (2) measles and adenoviral vectors can be constructed to target CD46 for cancer treatment. More than 20 CD46-targeted cancer treatments trials are currently being conducted.

**Figure 4 antibodies-09-00059-f004:**
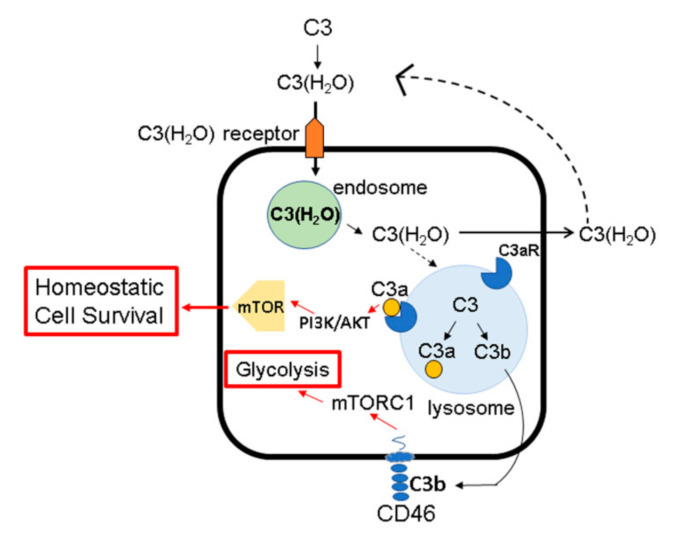
Intracellular C3 and CD46 drive cell metabolism and survival. Intracellular C3 is biosynthetically derived or can be loaded from the plasma. Generation of C3a and C3b from intracellular C3 can induce metabolic pathways through the PI3K-AKT-mTOR axis, enhancing glycolysis and homeostatic cell survival.

**Table 1 antibodies-09-00059-t001:** Clinical trials of CD46-based cancer therapeutics.

Name	Category	Target	Phase	Clinicaltrials.gov#	Sponsor
Enadenotucirev ^1^	Oncolytic adenovirus	Epithelial tumors; colorectal cancer; bladder cancer	Phase 1/2	NCT02028442	PsiOxus Therapeutics
Enadenotucirev	Oncolytic adenovirus + chemotherapy	Platinum-resistant epithelial ovarian cancer	Phase 1	NCT02028117	PsiOxus Therapeutics
Enadenotucirev	Oncolytic adenovirus + immunotherapy	Metastatic or advanced epithelial tumors	Phase 1	NCT02636036	PsiOxus Therapeutics
Enadenotucirev	Oncolytic adenovirus + chemoradiotherapy	Advanced rectal cancer	Phase 1	NCT03916510	University of Oxford
NG-350A ^2^	Oncolytic adenovirus	Advanced or metastatic epithelial tumors	Phase 1	NCT03852511	PsiOxus Therapeutics
NG-641 ^2^	Oncolytic adenovirus	Advanced or metastatic epithelial tumors	Phase 1	NCT04053283	PsiOxus Therapeutics
LOAd703 ^3^	Oncolytic adenovirus + chemotherapy or immunotherapy	Pancreatic cancer	Phase 1/2	NCT02705196	Lokon Pharma AB
LOAd703	Oncolytic adenovirus + immunotherapy	Malignant melanoma	Phase 1/2	NCT04123470	Lokon Pharma AB
LOAd703	Oncolytic adenovirus + chemotherapy or immune conditioning	Pancreatic adenocarcinoma; ovarian cancer; biliary cancer; colorectal cancer	Phase 1/2	NCT03225989	Lokon Pharma AB
MV-NIS ^4^	Oncolytic measles virus + chemotherapy	Cancer of ovaries, fallopian tubes or peritoneal cancer	Phase 2	NCT02364713	Mayo Clinic
MV-NIS	Oncolytic measles virus	Squamous cell carcinoma neck/head or breast cancer	Phase 1	NCT01846091	Mayo Clinic
MV-NIS	Oncolytic measles virus	Ovarian epithelial cancer; peritoneal cancer	Phase 1	NCT00408590	Mayo Clinic
MV-NIS	Oncolytic measles virus	Malignant pleural mesothelioma	Phase 1	NCT01503177	Mayo Clinic
MV-NIS	Oncolytic measles virus	Recurrent or refractory multiple myeloma	Phase 1/2	NCT00450814	Mayo Clinic
MV-NIS	Oncolytic measles virus	Peripheral nerve sheath tumor; neurofibromatosis type 1	Phase 1	NCT02700230	Mayo Clinic
MV-NIS	Oncolytic measles virus	Multiple myeloma	Phase 2	NCT02192775	University of Arkansas
MV-NIS	Oncolytic measlesvirus-infected mesenchymal stem cells	Recurrent ovarian cancer	Phase 1/2	NCT02068794	Mayo Clinic
MV-CEA ^5^	Oncolytic measles virus	Recurrent glioblastoma multiforme	Phase 1	NCT00390299	Mayo Clinic
MV-CEA	Oncolytic measles virus	Recurrent ovarian epithelial cancer;primary peritoneal cancer	Phase 1	NCT00408590	Mayo Clinic
TMV-018 ^6^	Oncolytic measles virus	Gastrointestinal cancer	Phase 1	NCT04195373	Themas Bioscience
FOR46 ^7^	Antibody-drug conjugate	Multiple myeloma, relapsed or refractory	Phase 1	NCT03650491	Fortis
FOR 46	Antibody-drug conjugate	Metastatic prostate cancer	Phase 1	NCT03575819	Fortis

^1^ Chimera derived from adenovirus group BAd11p/Ad3. ^2^ Transgene-modified variant of Enadenotucirev expressing a bi-specific T-cell activator molecule (FAP-TAc) recognizing fibroblast activating protein (FAP) on cancer associated fibroblasts (CAFs) and CD3 on T-cells. Production of FAP-TAc by virus-infected tumor cells should lead to T cell-mediated killing of CAFs and, thus, modification of the tumor microenvironment to drive effective anti-tumor immunity. ^3^ LOAd, the virus is a hybrid derived from adenovirus serotypes 5 and 35. It expresses immune-activating genes (trimerized membrane-bound isoleucine zipper TMZ-CD40L and 4-1BB ligand) under control of a cytomegalovirus (CMV) promoter. ^4^ MV-NIS, an oncolytic measles virus (MV) encoding the thyroidal sodium iodide symporter (NIS) that facilitates viral gene expression and offers a tool for radiovirotherapy. ^5^ MV-CEA, an oncolytic measles virus encoding the carcinoembryonic antigen (CEA). ^6^ Oncolytic measles virus, TMV-018-101, engineered with cytosine deaminase. ^7^ A monoclonal antibody to a conformationally-specific epitope of CD46 expressed only on tumor cells that is conjugated to an anti-cancer drug.

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
