# Peer review of "CD46 and Oncologic Interactions: Friendly Fire against Cancer"

_2073-4468, 2020, doi:10.3390/antib9040059_

Round 1

Reviewer 1 Report

The manuscript of Elvington et al addresses the role of CD46 in cancer. This is a timely and original overview of the subject, such review is lacking in the literature. It is well written and illustrated and easy to follow. One aspect which may be addressed with a small paragraph is the fact that according to TCGA database, in human solid tumors the overexpression of the CD46 gene is associated either with favorable (renal or stomach cancer) or poor prognosis (cervical cancer) or has no impact. This duality may be interesting to address with hypothesis for the potential mechanism of action in the different contexts.  

Author Response

We thank this reviewer for their thoughtful assessment of our manuscript. The point made about the variable course of CD46 expression was excellent. We have added a paragraph on page 5 at the end of section 2 discussing this. We appreciate the time and effort made to help us improve our manuscript.

Reviewer 2 Report

This is a well-written review.

I have only one cautionary comment regarding the over generalization of observations to multiple cancers in some-cases without a strong evidence. Authors may consider listing malignancies that targeting CD46 might be helpful based on evidence. One suggestion is to add a primary support literature and a line of explanation on listed clinical trials.

Author Response

We thank the reviewer for the assessment of our manuscript. In order to address this concern and better drive home the point of CD46 variable expression and its impact on cancers, we have added a paragraph on page 5 at the end of section 2 discussing this.

Reviewer 3 Report

In this review, the author has introduced CD46, which is highly overexpressed on a variety of human tumor cells, as a novel therapeutic target for cancer treatment. The author has provided biological background of CD46 and how CD46 plays a role in malignant transformation and cancer immunotherapy as a tumor target. They also introduced how CD46 can be used as a receptor for anti-tumor therapeutic vectors as well as a metabolic driver with a list of clinical examples. Overall I think this review is a comprehensive review for CD46’s application in oncologic connections. The paper is well-written and well-organized.

Author Response

We thank the reviewer for the very positive comments regarding our manuscript.  We appreciate the time and effort taken to review our work.